# An experimental machine learning study investigating the decision-making process of students and qualified radiographers when interpreting radiographic images

Clare Rainey[1]⊕*, Angelina T. Villikudathil[1]⊕, Jonathan McConnell[2], Ciara Hughes[1], Raymond Bond[3], Sonyia McFadden[1]

1 Faculty of Life and Health Sciences, School of Health Sciences, Ulster University, York Street, Belfast, Northern Ireland, United Kingdom, 2 NHS Leeds Teaching Hospitals, Leeds, United Kingdom, 3 Faculty of Computing, School of Computing, Engineering and the Built Environment, Ulster University, York Street, Belfast, Northern Ireland, United Kingdom

⊕ These authors contributed equally to this work.
* c.rainey@ulster.ac.uk

**Data Availability Statement:** Data cannot be shared publicly because of ethical restrictions on

## Abstract

AI is becoming more prevalent in healthcare and is predicted to be further integrated into workflows to ease the pressure on an already stretched service. The National Health Service in the UK has prioritised AI and Digital health as part of its Long-Term Plan. Few studies have examined the human interaction with such systems in healthcare, despite reports of biases being present with the use of AI in other technologically advanced fields, such as finance and aviation. Understanding is needed of how certain user characteristics may impact how radiographers engage with AI systems in use in the clinical setting to mitigate against problems before they arise. The aim of this study is to determine correlations of skills, confidence in AI and perceived knowledge amongst student and qualified radiographers in the UK healthcare system. A machine learning based AI model was built to predict if the interpreter was either a student (n = 67) or a qualified radiographer (n = 39) in advance, using important variables from a feature selection technique named Boruta. A survey, which required the participant to interpret a series of plain radiographic examinations with and without AI assistance, was created on the Qualtrics survey platform and promoted via social media (Twitter/LinkedIn), therefore adopting convenience, snowball sampling This survey was open to all UK radiographers, including students and retired radiographers. Pearson's correlation analysis revealed that males who were proficient in their profession were more likely than females to trust AI. Trust in AI was negatively correlated with age and with level of experience. A machine learning model was built, the best model predicted the image interpreter to be qualified radiographers with 0.93 area under curve and a prediction accuracy of 93%. Further testing in prospective validation cohorts using a larger sample size is required to determine the clinical utility of the proposed machine learning model.

sharing sensitive data. Access to the data can be provided following a successful application to Ulster University's Nursing and Health Research Ethics Filter Committee. Ulster University's Research Portal contains metadata on the dataset and instructions on how to request access to this dataset. This information can be accessed at https://doi.org/10.21251/50890091-4b54-4644-b980-ea9da646aa0e.

**Funding:** This work was supported by the College of Radiographers Research Industry Partnership award Scheme (CoRIPS: Grant number 213) (CR (P.I.), SMF, CH, RB, JMcC). The funders had no role in study design, data collection and analysis, decision to publish, or preparation of the manuscript.

**Competing interests:** The authors have declared that no competing interests exist.

## Author summary

Artificial Intelligence (AI) is becoming increasingly integrated into healthcare systems. Radiology, as a technologically advanced profession, is one area where AI has proved useful for myriad tasks. Developments in computer vision have allowed for applications in computer aided diagnosis using radiographic images. The integration and development of AI has been supported by healthcare providers and government agencies, such as the NHS in the UK. With the introduction of these systems in the clinical setting it is imperative to understand the nuances of the interaction between the clinicians using the technology and the system. Trust has been cited as a potentially significant issue in the effective integration of AI in radiology. Our research pinpoints the factors which have an impact on trust in radiographers and clarifies the strength of these associations. The means of analysis (Boruta) used provides an assessment of any correlations, allowing for intervention to be made. We found that females were less likely to trust AI that males and that trust in AI was negatively correlated with age and level of experience, indicating areas were further investigation and intervention are needed to ensure balanced trust and effective integration of AI in clinical radiography.

## Introduction

Artificial Intelligence systems are becoming more integrated into healthcare settings [1]. These systems have been proposed as a means to 'free up' clinicians' time, increase accuracy and reduce error [2], however, there have been few studies investigating the impact of these systems on the humans who are using them. Research has been conducted into the impact of computer assistance in other technologically advanced fields, such as aviation and the financial sector [3,4]. In healthcare, studies have focused on the impact of AI on decision making and Automation Bias (AB) [5,6]. These studies have highlighted some of the potential problems which may become more prevalent with the growing integration of AI systems in healthcare, however further exploration into other factors which may impact human behaviour when using computer assisted AI is lacking.

Decision support systems and AI are being developed in conjunction with clinicians in many areas of medicine. Recent developments in computer vision have permitted AI systems to be further developed for use in radiology, with promising results reported in many studies [7,8,9,10]. Neural networks are commonly used in computer vision tasks. This type of system comes with its own unique challenges, due in large part to the mode of operation, where some of the processing takes place in 'hidden layers', the so-called 'black box' of AI. This has led to a lack of trust from clinicians, who are expected to assume ultimate responsibility for the eventual diagnosis, whether assisted by AI or not.

The rate of the development of AI systems for use in healthcare has been increasing rapidly with many systems already in place [1]. Understanding of the human interaction with such systems will permit a critical approach and allow for an awareness of the characteristics of those who may be more susceptible to either over or under reliance on such technologies.

Increasing pressure on healthcare systems may provide much of the motivation for further integration of AI in medicine [11,12,13,14,15,16,17]. Staff shortages, coupled with increased demand for services mean that many countries are training and employing new staff in order to fill the backlog. Previous studies have found that less experienced clinicians may be more likely to trust computer assisted AI, inducing errors of both commission and omission [5,6,18,19]. Reliance on the AI diagnosis can be beneficial when the AI feedback is correct but

can be detrimental when error exists in the advice given. Conversely, experienced clinicians report a lack of trust in AI systems and may, therefore, not derive the benefit of the assistance, choosing to rely on their own knowledge and experience. Hence, this study sought to identify potential user characteristics which might predispose them to certain behaviours when using AI e.g., lack of trust.

This paper answers the following research questions in relation to clinicians interacting with AI:

- What are the clinical variables (features) that are significant when predicting student and qualified radiographers' interactions.

- What is the best predictive performance of a machine learning (ML) model in predicting performance of student and qualified radiographers.

- What are the clinical correlations between student and qualified radiographers.

## Using ML to discover knowledge in a radiographer dataset

Models built using ML algorithms require constant monitoring of their predictive performance when used in prediction or classification tasks [20] due to the probabilistic nature of risk prediction tools [21]. Monitoring tests such as stability of features are crucial in ML. There are different ML algorithms to address different types of analytical problems: from detection, localisation, assessment to prediction, classification, regression or density estimation tasks [22]. The predictive performance of a ML model can be enhanced by investigating the features that drive the prediction task [23]. This can lead to knowledge discovery, especially when we identify features or variables that stand out as important in prediction and classification tasks.

To assess the performance of the ML models, feature selection is often used. In most bioinformatics analysis [24], feature selection has become a pre-requisite for model building. Feature selection techniques select a subset of variables; in the case of projection, principal component analysis is evident and in the case of compression, information theory is used. The main objectives of feature selection are: a) To select the most predictive features, b) To improve model performance and to avoid over-fitting, and c) To provide more cost-effective and faster models.

There are three main types of feature selection methods, two of which are used here: (i) The filter-based feature selection method looks at the relevance of the features by using a scoring system. For example, Chi-square test is a non-parametric statistical test that assesses the dependence between the class and the feature as a measure of Chi-square statistics [25]. (ii) The wrapper-based feature selection method looks for a specific subset of features by training and testing the classification algorithm. For example, Boruta is a wrapper-based ML algorithm that calculates and displays the Z-score distribution of the input features [26].

## Results

The correlation between continuous variables for all the participants of this study were investigated. Males who perceived themselves as proficient in their profession were more likely to trust AI as the male gender positively correlates with trust and with proficiency with a correlation coefficient (correlation = 0.67) >0. Trust was found to be negatively correlated with age and with level of experience with a correlation coefficient (correlation = -0.67 and -0.82 respectively) <0, meaning that the younger, less experienced participants reported less trust in AI. Age positively correlated with level of experience, i.e., the older respondents had more clinical experience. Age negatively correlated with practice, trust, males and proficiency, i.e., the

younger participants were found to have less trust in AI, were more often females and reported themselves to be less professionally proficient. Based on work location of the country where the participants were located, females who had less practice were less likely to trust AI as the work location negatively correlates with female gender, trust and practice with a correlation coefficient (correlation = -0.42, -0.34 and -0.34, respectively) <0, meaning that there may be locations which have a great population of female radiographers and that there exists a variability of trust in AI in different geographical regions. However, we do note that the males and female ratio relative to average workforce hasn't been recorded in our study. The variables: work location, age and level of experience were clustered together into one subgroup, while practice, trust, males, and proficiency were clustered together into another subgroup. (See Fig 1)

## Further analysis

compared the importance of continuous and categorical clinical variables for predicting whether the interpreter was a student or qualified radiographers.

The distribution of Z-scores boxplots were ranked by the Boruta algorithm and revealed level of experience, age and work location to be the significant features (above the shadow max attribute). See Fig 2

## High performing ML models useful to predict student and qualified radiographers

Five ML algorithms (Support Vector Machines, Naïve Bayes, Logistic Regression, k-Nearest Neighbour and Random Forest) were trained and tested for prediction of student and qualified radiographer groups. A detailed overview of the five comparative ML models' performance evaluation metrics, consisting of Area Under the Receiver Operating Characteristic Curve (AUC), Classification Accuracy (CA), Mathews Correlation Coefficient (MCC), precision (sensitivity), recall (specificity) and F1 score built for predicting students and qualified are represented in Table 1. When classifying the student and qualified radiographers the Naïve Bayes model shows maximal sensitivity of 93%, specificity of 93%, a MCC score of 0.85 and an AUC of 0.93; outperforming other ML models in Table 1. The Boruta significant variables from Fig 2 were used to build these ML models.

ROC curve comparisons of the top ML algorithms using significant Boruta variables were performed for student versus qualified radiographers. The x-axis in Fig 3 denotes False Positive Rate (FPR) prediction and y-axis denotes True Positive Rate (TPR) prediction. The dotted lines in the figures represents the ROC curve for a random classification model (random performance). Legend denotes the Area Under Curve (AUC) values obtained with different ML algorithms colour coded for differentiation.

## Discussion

In the present study, we have investigated the correlations of continuous variables for all the participants of this study using a novel ML approach for this task. Previous studies have identified the negative correlation between trust and age [27].

Males who were proficient in their profession were likely to trust AI. Literature suggests that in a male dominated respondents of a study, their trust level was statistically lower in usage of AI applications [28]. However, other studies have reported that radiologists who were more trained and informed in AI and have a higher trust in AI would be among those who would have higher adoption probability of AI applications [29]. This corroborates with our findings that males who were more proficient in their profession would likely trust AI.

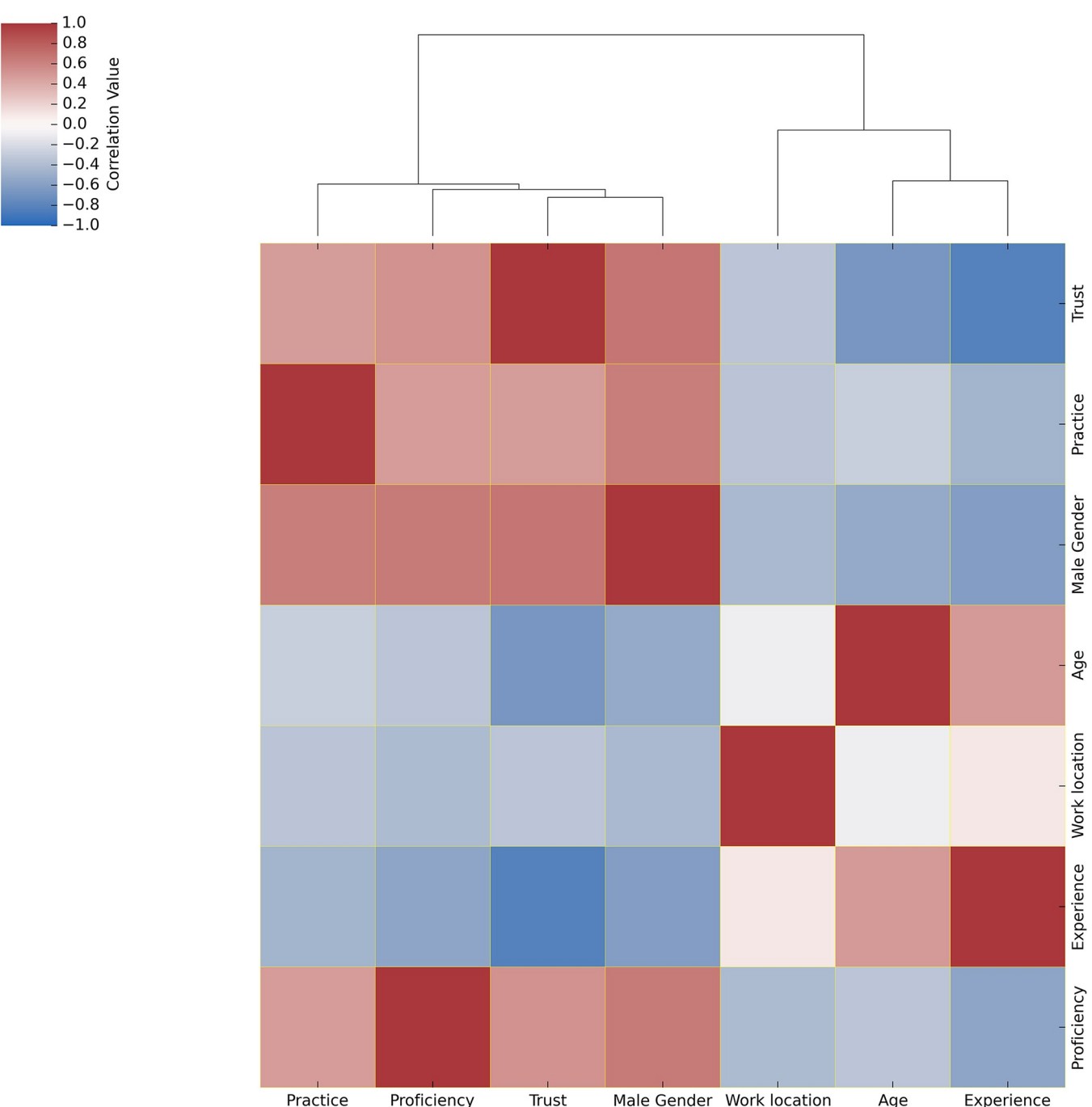

**Fig 1. Correlation heatmap of continuous variables for the entire participants of this study.** The continuous variables were imputed, normalised and Pearson's correlation was calculated. The resulting correlation matrix is plotted for all the participants of this study (n = 106). The colour scheme key ranging from 1 to -1 denotes red colour to be positively correlated, blue to be negatively correlated and white to have no correlation among the variables. Correlation clustering dendrogram depicts the relative associations among variables.

In our findings, trust was negatively correlated with age and with level of experience. This could most likely be the case because expert radiologists would likely trust and rely on their own skills instead of trusting AI for applications [30]. This is supported by existing literature on the subject indicating that more experienced professionals are less likely to show preference

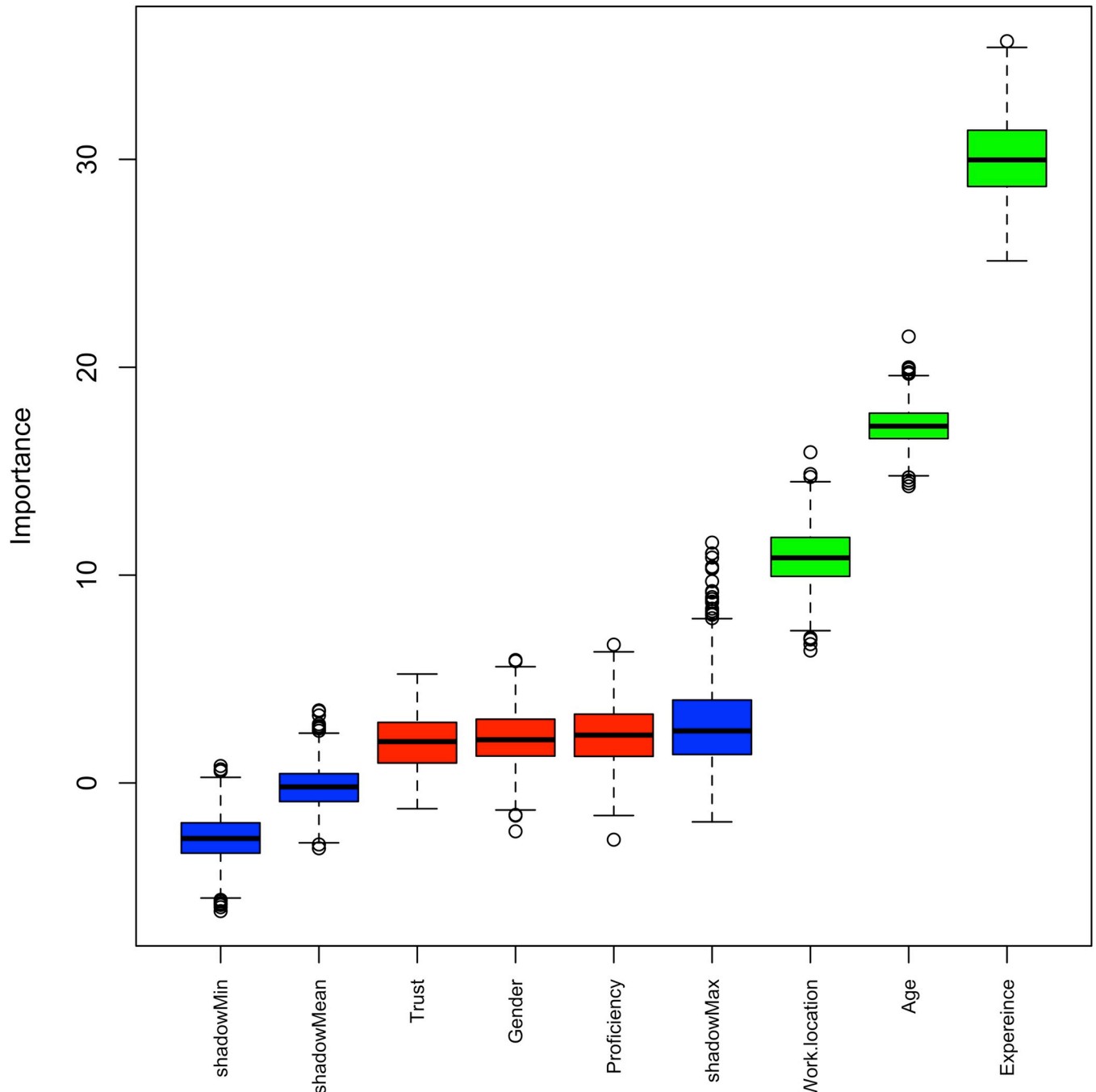

**Fig 2. Ranking of clinical variables using Boruta algorithm upon comparing student radiographers and qualified radiographers as participants of this study.** The variables were imputed, scaled, split based on training and testing of radiographic images reviewed, and Boruta algorithm was applied comparing students (n = 52) and qualified (n = 42) participants of this study. The resulting important variables are depicted as Z-score boxplots ranked by the Boruta algorithm wherein green colour denotes passed important variables, red denotes failed variables and blue denotes shadow (random variable) min, mean and max. Here, Fig 2 highlights level of experience, age and work location as important variables ranked by the Boruta algorithm.

for the input of AI in their diagnosis [5,6]. This is despite radiographers being accustomed to, and rely on, the use of technology in their day-to-day work. Furthermore, age was positively correlated with level of experience and negatively correlated with practice, trust, male and proficiency. Age and level of experience were found to be positively correlated, indicating that

**Table 1. Evaluation metrics of the ML models performance built for student and qualified radiographer groups.** Results are based on an average of the 3-fold cross validation. The top performing ML model and their metrics are highlighted for the comparison. SVM denotes for Support Vector Machines, NB for Naive Bayes, k-NN for K-Nearest Neighbour, LR for Logistic Regression, RF for Random Forest.

| Comparison type | ML Model | Area Under Curve (AUC) | Classification Accuracy (CA) | Matthew's Correlation Coefficient (MCC) | Positive Predictive Value (PPV) | Negative Predictive Value (NPV) | F1 score |
|---|---|---|---|---|---|---|---|
| Student versus qualified radiographers | SVM | 0.91 | 92.09±3.01% | 0.83 | 0.92 | 0.91 | 0.91 |
| | NB | 0.93 | 93.43±3.51% | 0.85 | 0.93 | 0.93 | 0.92 |
| | K-NN | 0.91 | 92.09±3.01% | 0.83 | 0.92 | 0.91 | 0.91 |
| | LR | 0.91 | 92.09±3.01% | 0.83 | 0.92 | 0.91 | 0.91 |
| | RF | 0.91 | 92.09±3.01% | 0.83 | 0.92 | 0.91 | 0.91 |

there is a greater proportion of young, newly qualified radiographers included in this study, rather than professionals who have entered the profession as mature students, as expert radiographers are more confident in AI in general with limitations in the ability to explain AI terminologies. However, younger radiographers miss out on the benefits of AI due to their inexperience and age.

Work location negatively correlated with females, trust and practice. The literature review suggests that female radiographers were likely to be unaware of using AI in clinical practice, due to their lack of trust in AI. Mistrust may be also due to the sparce uptake of technology in the UK healthcare system, despite of funding and focus from the government through the NHS [31]. Our findings indicate that female radiographers are less likely to use a highly complex AI system due to their lower trust.

The variables work location, age and level of experience were clustered together into one subgroup, while practice specialism ('practice'), trust, males and proficiency were clustered

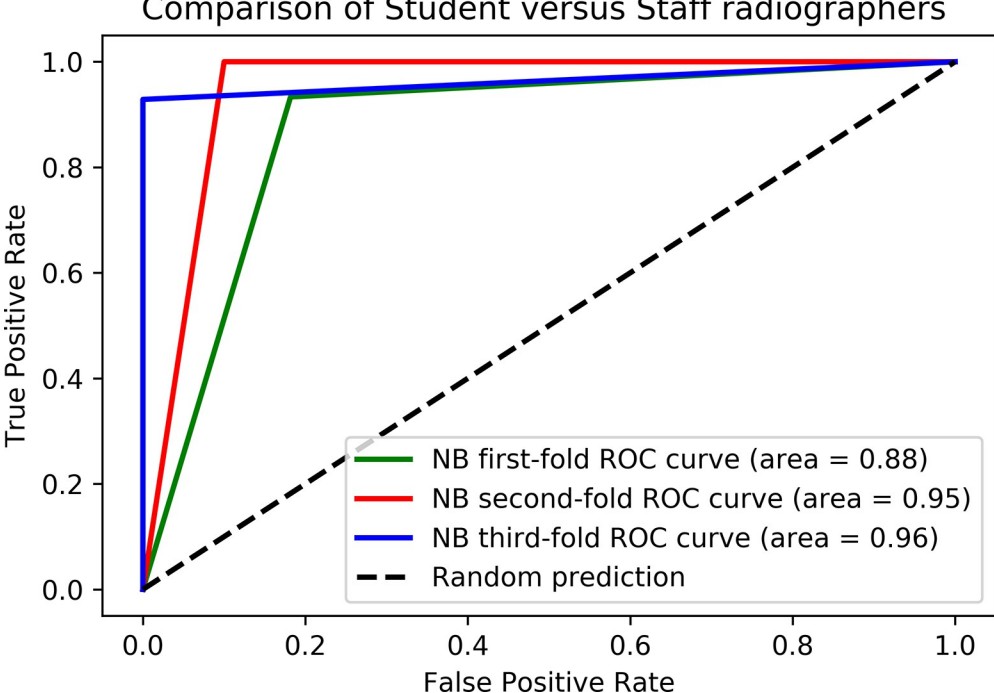

**Fig 3. Receiver Operator Characteristics (ROC) curves of the best performing machine learning algorithms using the Boruta important variables between student and qualified radiographer groups.**

together into another subgroup. The clustering of variables indicates similarity among variables.

The present results of ML AI predictive models are significant in at least two major respects. Firstly, the ML model built using algorithm Naive Bayes, has an excellent prediction accuracy of 93% and ROC curve area of 0.93 using the important Boruta ranked variables from student versus qualified radiographers. Furthermore, this approach may be used to test prospectively in clinical settings to predict qualified radiographer status, potentially earlier than is presently possible. The ML models built using the important Boruta ranked variables from student versus qualified radiographers can be improvised further with larger sample size and experimenting with other feature selection methods. Secondly, the identified important Boruta ranked variables can aid knowledge discovery and can be studied further in research settings. So far, ours is the first research design to have employed the usage of traditional ML based algorithms to predict student versus qualified radiographers using continuous variables. With the increasing use of new technologies in the clinical setting a simple and reliable means to identify any important corelations in how the human is predicted to interact with the system, and indeed, how the interaction may result in a negative patient outcome, will be useful to target education and training interventions. This may mean that those who are engaged as clinical experts in the new systems, as suggested by Rainey et al. (2022) and Strohm et al. (2020) [32,33] can have a more detailed knowledge of the specific intervention and support needed, both before implementation and during clinical use.

## Materials and methods

### Data collection and sample size of this study

Participants data were collected via the Qualtrics survey platform. The study was promoted via Twitter and LinkedIn to student, retired and practising radiographers. As a result of this, a total of 106 participants' data was collected (students n = 67, radiographers n = 39). Participants were asked to provide binary diagnosis (pathology/no pathology) on three, randomly allocated, plain radiographic examinations from a dataset of 21 full examinations. Each image was presented with and without AI assistance in the form of heatmaps (Grad-CAM) and binary diagnosis. The AI model was based on a ResNet-152 architecture, with the arithmetic mean of the output of the AI used to determine pathology (threshold 0.5). Preliminary findings from this initial study were presented at the European Congress of Radiology [34].

The study was designed by the authors of this paper and piloted on a group of experienced radiographers (n = 3) and student radiographers from each year group of the Diagnostic Radiography and Imaging programme at Ulster University (n = 3). Face and content validity were ensured by request for feedback on the accessibility of the platform and comprehension of the content. Additionally, participants were asked to indicate the time commitment required to complete the survey. Three examinations were chosen as this was deemed an acceptable time sacrifice, supported by other studies [35]. No changes were made as a result of the pilot study as all participants were content that the survey was accessible, understood and measured the aim of the research.

The radiographic images used in the study were previously used in other studies [36]. The sim of the initial study was to clarify the impact of different forms of AI on student and qualified radiographers' interaction with the technology.

Demographic information such as age, level of experience, work location, area of practice (or specialism) and proficiency in the use of technology in day-to-day life was collected. Trust in AI was determined on a scale of 0 to 5 ranging from not at all to absolute trust in AI for each examination. The survey was openly available from 2nd March to 2nd November 2021.

## Correlation analysis

Pearson's correlation coefficient was used to find correlations between the variables, and these are presented as heatmaps. Pearson's correlation coefficient was also used to determine the direction and strength of association between variables; it measures if any linear association could occur between the variables under study [37]. Consequently, the variables (continuous and categoric) that were used in the correlation analysis were imputed using simple imputer function with the mean of the column, normalised using normalizer function and Pearson's correlation matrix was calculated for all the participants of this study. Results were presented as heatmaps.

## Clustering analysis

The Euclidean distance between each variable was computed, followed by hierarchical clustering, to check for any specific sub-clusters of participants in this study. A Euclidean distance metric was used to determine the distance between an existing datapoint and a new data point [37], which forms the basis of measure of similarity and dissimilarity between the two data points. Consequently, the continuous clinical variables and the categorical variables were imputed using simple imputer function, normalised using normalizer function with the mean of the column and Euclidean distance metric computed for all the participants of this study. Results were presented as heatmaps.

## Boruta feature selection

The Boruta algorithm in R interface, which is a Random Forest based wrapper method, was used for feature selection [26]. Here, continuous and categorical variables were imputed using MICE imputation function for all the missing data, split based on 80% training and 20% test set. MICE imputation performs multiple regressions on random samples of the data and aggregates for imputing the missing values. Scaling of variables was performed using standard scaler function and the Boruta algorithm was applied for all the participants of this study.

## Machine learning model development and evaluation

The Boruta significant variables were used as input features to develop five ML based models using following algorithms: Support Vector Machine (SVM), Naive Bayes (NB), K-Nearest Neighbour (K-NN), Logistic Regression (LR) and Random Forest (RF). The comparative model performance was assessed by Receiver Operator Characteristic (ROC) curves. Here, the gold standard refers to the most accurate and reliable method available for determining the student versus qualified radiographers. The ML models were built to assess the efficiency of the Boruta significant variables between student and qualified radiographers. Cross validation was used to enable a robust estimation of the performance of the ML model. Here, the ML models were built using a 3-fold stratified split cross validation and they were assessed based on their predictive performance using model evaluation metrics such as Area Under Curve (AUC), Classification Accuracy (CA), Matthew's Correlation Coefficient (MCC), Positive Predictive Value (PPV), Negative Predictive Value (NPV) and F1 score.

To enable a robust estimation of the performance of a ML model, cross validation is performed. Depending upon the size of the dataset, different types of cross validation methods can be used, e.g. $n$-fold, random sampling, leave-one-out, etc. In the presented work, an $n$-fold cross validation method was performed, wherein at each step of cross validation, $n$-1 (i.e. total number of subsets—1) subsets are merged as the training set to train the model; while the

remaining 1 subset is used as the test set to validate the trained model. This procedure is repeated *n* times such that each subset is used in training and then validated in testing [38].

Several metrics are used to evaluate the performance of a ML based model. The commonly used metrics for ML model evaluation are:

i.   Positive Predictive Value (PPV): This test identifies the proportion of true positives out of the total of true (TP) and false (FP) positives [39].
     PPV = TP / TP+FP

ii.  Negative Predictive Value (NPV): This test identifies the proportion of true negatives out of the total of true (TN) and false (FN) negatives [39].
     NPV = TN / TN+FN

iii. Classification Accuracy (CA): This determines the overall predictive accuracy of the ML model that has been trained [40].
     CA = TP+TN / TP+TN+FP+FN

iv.  F1 score: Measures the accuracy of a test by calculating the true positives to the arithmetic mean of real positives (precision) and predicted positives (recall), wherein F1 score of 1 depicts best accuracy and 0 represents lowest accuracy [41].
     F1 = (TN/(TN+FP)x (TP/(TP+FN))/(TN/(TN+FP))+(TP/(TP+FN))

v.   Mathews Correlation Coefficient (MCC): It is widely used as a performance measure to validate predictive models. The MCC metric calculation uses four quantities (TP, TN, FP, FN) and gives in score ranging from -1 to 1, with 1 depicting complete agreement, -1 depicting complete disagreement and 0 depicting that the prediction is uncorrelated [42].
     MCC = ((TPxTN)(FPxFN))/$\sqrt{}$((TP+FP)x(TP+FN)x(TN+FP)x(TN+FN))

vi.  Area Under Curve (AUC): The area under the Receiver Operating Characteristic (ROC) curve helps to compare the performance of different ML classifier models [43].
     TPR = ∫ (T to ∞) f1(x)dx; FPR = ∫ (T to ∞) f0(x)dx; Area = $P(X_1 > X_0)$; where $X_1$ is the score for positive instance and $X_0$ is the score for negative instance.

## Usage of software and libraries

All analyses were performed in Jupyter Notebook [44] using python [45] Version 2 and packages: pandas [46]; numpy [47]; SimpleImputer [48]; Normalizer [48]; seaborn [49]; matplotlib [50]; math [51]; mice [52]; train_test_split [53]; StandardScaler [54]; DataFrame [55]; and RStudio [56] Version 2022 and statistical functions and libraries: ranger [43] and Boruta package [57] was used for feature importance ranking and feature selection.

## Acknowledgments

Ethical permission for this study has been granted by Ulster University Nursing and Health Research Ethics Filter Committee FCNUR-20-035. Ethical permission for the use of the clinical dataset images has been granted previously to use the images for research purposes (Monash University, Clayton, Australia, 2011).

## Author Contributions

**Conceptualization:** Clare Rainey, Angelina T. Villikudathil, Sonyia McFadden.

**Data curation:** Clare Rainey, Angelina T. Villikudathil, Sonyia McFadden.

**Formal analysis:** Angelina T. Villikudathil.

**Funding acquisition:** Clare Rainey, Sonyia McFadden.

**Investigation:** Clare Rainey, Angelina T. Villikudathil, Ciara Hughes, Raymond Bond, Sonyia McFadden.

**Methodology:** Clare Rainey, Angelina T. Villikudathil, Sonyia McFadden.

**Project administration:** Clare Rainey, Ciara Hughes, Raymond Bond, Sonyia McFadden.

**Resources:** Clare Rainey, Sonyia McFadden.

**Software:** Clare Rainey, Sonyia McFadden.

**Supervision:** Clare Rainey, Jonathan McConnell, Ciara Hughes, Raymond Bond, Sonyia McFadden.

**Validation:** Clare Rainey, Jonathan McConnell, Ciara Hughes, Raymond Bond, Sonyia McFadden.

**Visualization:** Clare Rainey, Sonyia McFadden.

**Writing – original draft:** Clare Rainey, Angelina T. Villikudathil.

**Writing – review & editing:** Clare Rainey, Angelina T. Villikudathil, Jonathan McConnell, Ciara Hughes, Raymond Bond, Sonyia McFadden.

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
