## [Decision Letter · Decision Letter 0]

13 Jul 2023

PDIG-D-23-00083

An experimental machine learning study investigating the decision-making process of students and qualified radiographers when interpreting radiographic images.

PLOS Digital Health

Dear Dr. Rainey,

Thank you for submitting your manuscript to PLOS Digital Health. After careful consideration, we feel that it has merit but does not fully meet PLOS Digital Health's publication criteria as it currently stands. Therefore, we invite you to submit a revised version of the manuscript that addresses the points raised during the review process.

Please submit your revised manuscript within 30 days Aug 12 2023 11:59PM. If you will need more time than this to complete your revisions, please reply to this message or contact the journal office at digitalhealth@plos.org. Please include the following items when submitting your revised manuscript:

We look forward to receiving your revised manuscript.

Kind regards,

Ana Luísa Neves

Academic Editor

PLOS Digital Health

Journal Requirements:

2. We ask that a manuscript source file is provided at Revision. Please upload your manuscript file as a .doc, .docx, .rtf or .tex.

Additional Editor Comments (if provided):

Reviewers' comments:

Reviewer's Responses to Questions

**Comments to the Author**

1. Does this manuscript meet PLOS Digital Health’s publication criteria? Is the manuscript technically sound, and do the data support the conclusions? The manuscript must describe methodologically and ethically rigorous research with conclusions that are appropriately drawn based on the data presented.

Reviewer #1: Yes

Reviewer #2: Yes

2. Has the statistical analysis been performed appropriately and rigorously?

Reviewer #1: Yes

Reviewer #2: Yes

3. Have the authors made all data underlying the findings in their manuscript fully available (please refer to the Data Availability Statement at the start of the manuscript PDF file)?

Reviewer #1: Yes

Reviewer #2: Yes

4. Is the manuscript presented in an intelligible fashion and written in standard English?

Reviewer #1: Yes

Reviewer #2: Yes

5. Review Comments to the Author

Reviewer #1: Abstract

Here we report analysis from 106 student and qualified radiographers….revise this to indicate how many students and qualified radiographers.

A brief regarding the technical operation of the AI tool developed and employed here is critical.

The language and structure of the abstract could be improved with emphasis on the key implication of the findings to current clinical practice, clear separation between intro/background, methods, results, conclusions. 

Introduction

….. with promising results reported in many studies. Which studies??

The introduction seem too long and this could be summarised to present the message in a clearer and succinct form.

Key relevant literatures in the area of previous reports on the workforce under consideration are missing from the introduction and/or discussion: Few that come to mind for your consideration: Malamateniou et al (Radiography); Botwe et al (Radiography); Botwe et al (JMRS); Antwi et al (Insights into Imaging): Malamateniou et al and Akudjedu et al (all in JMIRS). Some of these are of global relevance and thus, these discussions/perspectives are critical.

Methods

Was the study approved by any ethics panel?

How long was the online survey live for? Any reminders? 

How was the survey instrument designed? 

Was Piloting of the tool done and what was the result?

 Pre-test of the survey tool to ensure this was compliant across a range of platforms (e.g.,android vrs iOS, Mac vrs MS, computer vrs mobile etc)

Results

Good – thank you

Discussion

The clinical applicability and implications could be discussed a bit more in-depth, this is what will be of interest to readers.

`what are the study limitations ? 

General Comments

Good work – however, please pay attention to grammar and language – at some points there is a mix of both American and British English.

Reviewer #2: Manuscript Number: PDIG-D-23-00083

Title: An experimental machine learning study investigating the decision-making process of

students and qualified radiographers when interpreting radiographic images.

Abstract: Well written, concise and provides the required information about the study.

• In the following: ‘Here we report analysis from 106 participants: students (n=?) and qualified radiographers (n=?) as participants of an experimental study’.

o Indicate how many students and how many qualified radiographers.

Introduction: 

• Provides a good overview of the current situation and justification of the study.

Results: 

• On 1st reference of the area under the curve (AUC), there is no reference to the RoC curve.

o What was the gold standard used for this RoC study (include in methodology section)?

Discussion: 

• Good discussion of study findings backed up and compared to literature.

Materials and Methods: 

• There is a lack of information about the methodology used in conducting the study. In the abstract reference is made to a survey but no details given. In the results, there is reference to interviews. Clarification required.

• What were participants asked to do? Review a set of images?

o Details pertaining to the above is lacking.

6. PLOS authors have the option to publish the peer review history of their article (what does this mean?). If published, this will include your full peer review and any attached files.

**Do you want your identity to be public for this peer review?** For information about this choice, including consent withdrawal, please see our Privacy Policy.

Reviewer #1: No

Reviewer #2: Yes: Prof Francis Zarb

---

## [Decision Letter · Decision Letter 1]

29 Jul 2023

An experimental machine learning study investigating the decision-making process of students and qualified radiographers when interpreting radiographic images.

PDIG-D-23-00083R1

Dear Mrs Rainey,

We are pleased to inform you that your manuscript 'An experimental machine learning study investigating the decision-making process of students and qualified radiographers when interpreting radiographic images.' has been provisionally accepted for publication in PLOS Digital Health.

Best regards,

Ana Luísa Neves

Academic Editor

PLOS Digital Health

Reviewer Comments (if any, and for reference):

Reviewer's Responses to Questions

**Comments to the Author**

1. If the authors have adequately addressed your comments raised in a previous round of review and you feel that this manuscript is now acceptable for publication, you may indicate that here to bypass the “Comments to the Author” section, enter your conflict of interest statement in the “Confidential to Editor” section, and submit your "Accept" recommendation.

Reviewer #1: All comments have been addressed

Reviewer #2: All comments have been addressed

2. Does this manuscript meet PLOS Digital Health’s publication criteria? Is the manuscript technically sound, and do the data support the conclusions? The manuscript must describe methodologically and ethically rigorous research with conclusions that are appropriately drawn based on the data presented.

Reviewer #1: Yes

Reviewer #2: (No Response)

3. Has the statistical analysis been performed appropriately and rigorously?

Reviewer #1: Yes

Reviewer #2: (No Response)

4. Have the authors made all data underlying the findings in their manuscript fully available (please refer to the Data Availability Statement at the start of the manuscript PDF file)?

Reviewer #1: No

Reviewer #2: (No Response)

5. Is the manuscript presented in an intelligible fashion and written in standard English?

Reviewer #1: Yes

Reviewer #2: (No Response)

6. Review Comments to the Author

Reviewer #1: Thank you for addressing all my comments - best of luck!

Reviewer #2: (No Response)

7. PLOS authors have the option to publish the peer review history of their article (what does this mean?). If published, this will include your full peer review and any attached files.

**Do you want your identity to be public for this peer review?** For information about this choice, including consent withdrawal, please see our Privacy Policy.

Reviewer #1: **Yes: **Dr Theo Akudjedu

Reviewer #2: None
